# The Presence of Pre-Existing Endometriotic Lesions Promotes the Growth of New Lesions in the Peritoneal Cavity

**DOI:** 10.3390/ijms241813858

**Published:** 2023-09-08

**Authors:** Ilinca T. Mihai, Jeannette Rudzitis-Auth, Michael D. Menger, Matthias W. Laschke

**Affiliations:** Institute for Clinical and Experimental Surgery, Saarland University, 66421 Homburg, Germany; ilincateodoramihai@gmail.com (I.T.M.); jeannette.rudzitis-auth@uks.eu (J.R.-A.); michael.menger@uks.eu (M.D.M.)

**Keywords:** endometriosis, ultrasound, bioluminescence, mice, proliferation, inflammation, immune cells, vascularization

## Abstract

Endometriosis is a common gynecological disease which is characterized by endometriotic lesions outside the uterine cavity. In this study, we investigated whether the presence of pre-existing endometriotic lesions promotes the development of new lesions due to the exchange of cells and an altered peritoneal environment. For this purpose, uterine tissue samples from FVB/N wild-type donor mice were transplanted simultaneously or time-delayed with samples from transgenic FVB-Tg(CAG-luc-GFP)L2G85Chco/J donor mice into the abdominal cavity of FVB/N wild-type recipient mice. The formation of endometriotic lesions was analyzed by means of high-resolution ultrasound, bioluminescence imaging, histology and immunohistochemistry. Moreover, immune cells and inflammatory factors in the peritoneal fluid were assessed by flow cytometry and a cytokine array. These analyses revealed that the growth of newly developing endometriotic lesions is promoted by the presence of pre-existing ones. This is not due to an exchange of cells between both lesion types but rather caused by peritoneal inflammation induced by already established lesions. These findings indicate that, among other pathogenic mechanisms, the chronic nature of endometriosis may be driven by a lesion-induced inflammatory milieu in the peritoneal cavity, which creates favorable conditions for the development of new lesions.

## 1. Introduction

Endometriosis is one of the most common benign diseases in gynecology, affecting approximately 16–20% of all women of reproductive age worldwide [1]. Patients often suffer from dysmenorrhea, dyspareunia, chronic lower abdominal pain and infertility [2]. Due to frequent hospitalization, numerous surgeries and expensive medical treatment, as well as recurrence rates of up to 80%, the disease has a considerable economic impact on the health care system [3]. In addition, endometriosis markedly affects the patients’ quality of life [4].

Endometriosis is defined by the ectopic growth of endometrial glandular and stromal cells as endometriotic lesions outside the uterine cavity. A possible explanation for the development of these lesions is given by the implantation theory of Sampson [5]. He postulated that during menstruation, vital endometrium is retrogradely shed from the uterus through the fallopian tubes into the abdomen, where it adheres to the peritoneum and develops into vascularized endometriotic lesions [6]. This process may markedly alter the physiological microenvironment in the abdominal cavity. In line with this view, the peritoneal fluid of endometriosis patients has been shown to contain high concentrations of various inflammatory cytokines and angiogenic growth factors [7,8]. In this study, we hypothesized that these microenvironmental changes caused by pre-existing lesions may promote the development of new endometriotic lesions in the sense of a vicious circle. Moreover, it has recently been demonstrated that green fluorescent protein (GFP)-positive cells from the uterus incorporate into murine endometriotic lesions [9]. Hence, it may be assumed that lesion formation is further supported by an exchange of cells between individual lesions.

To test these hypotheses, we transplanted uterine tissue samples from FVB/N wild-type donor mice simultaneously or time-delayed with samples from transgenic FVB-Tg(CAG-luc-GFP)L2G85Chco/J donor mice into the abdominal cavity of FVB/N wild-type recipient mice for the surgical induction of endometriotic lesions. In the used transgenic mouse line, all cells express firefly luciferase directed by the CAG promoter (human cytomegalovirus immediate early promoter enhancer with chicken beta-actin/rabbit beta-globin hybrid promoter). Accordingly, the cross-over design of our transplantation experiments allowed the analysis of a possible recruitment of luciferase-positive cells in lesions developing from uterine tissue samples of FVB/N wild-type mice using bioluminescence imaging. In addition, we analyzed the growth, proliferation, vascularization and immune cell infiltration of newly forming endometriotic lesions by means of high-resolution ultrasound, histology and immunohistochemistry. Furthermore, immune cells and inflammatory factors in the peritoneal fluid were assessed by flow cytometry and a cytokine array.

## 2. Results

### 2.1. Development and Growth of Endometriotic Lesions

In a first set of experiments, 16 uterine tissue samples from six transgenic FVB-Tg(CAG-luc-GFP)L2G85Chco/J mice and 16 uterine tissue samples from six FVB/N wild-type mice were simultaneously transplanted (ST) on day 0 (d0) into the abdominal cavity of eight wild-type mice (Figure 1A). Two luciferase-positive and two luciferase-negative samples per animal were sutured at defined locations to the peritoneum at each side of the abdominal wall. In a second set of experiments, a time-delayed transplantation (DT) of the two different types of uterine tissue samples was performed (Figure 1B).

For this purpose, a total of 16 uterine tissue samples from two transgenic mice were first transplanted to the right abdominal wall of eight recipient wild-type mice (d-14). After two weeks (d0), the animals were laparotomized again and 16 uterine tissue samples from two wild-type mice were additionally transplanted to the left side of the abdominal wall. Only the uterine tissue samples from the wild-type mice were analyzed by means of repeated high-resolution ultrasound imaging over an observation period of 28 days. Subsequently, the samples from both the wild-type and transgenic mice were excised and processed for histological analyses. Hematoxylin and eosin (HE) stainings confirmed that in each animal at least one sample of wild-type and transgenic origin had finally developed into an endometriotic lesion, as defined by the presence of endometrial stroma and glands (Figure 2A,B). Only these lesions, but not completely regressed grafts, were included in further analyses. 

High-resolution ultrasound imaging (Figure 2C–J) revealed a comparable initial volume of wild-type lesions in the ST and DT group, indicating standardized baseline conditions (Figure 2E). However, during the following observation period of 28 days, only the lesions of the DT group progressively grew and exhibited a significantly increased lesion volume as well as stromal tissue and cyst volume when compared to d0 (Figure 2E,G,I). Accordingly, the growth rate of their overall volume and stromal tissue volume was also higher when compared to the lesions of the ST group (Figure 2F,H). In contrast, the lesions of the ST group did not markedly grow and, thus, presented a significantly lower overall volume and a stromal tissue volume between d7-d28 when compared to lesions of the DT group (Figure 2E,G). Although the number of cyst-containing lesions increased in the DT group over time, a comparable cyst volume between the two groups was detected (Figure 2I,J).

### 2.2. Exchange of Cells between Endometriotic Lesions

The exchange of cells between endometriotic lesions was analyzed by means of bioluminescence imaging. To verify the appropriateness of our experimental setting, we first performed bioluminescence imaging of the FVB-Tg(CAG-luc-GFP)L2G85Chco/J and FVB/N wild-type mice as well as their uterine horns and uterine tissue samples with and without the administration of synthetic D-luciferin (Figure 3A,B). This analysis revealed that luciferase is strongly expressed in the entire body of the transgenic mice. As expected, no signals were detected in transgenic mice without luciferin administration and in FVB/N wild-type mice with injected luciferin (Figure 3A). Comparable results were found for the ex vivo analyzed uterine horns and uterine tissue samples (Figure 3B).

After the induction of endometriotic lesions, bioluminescence imaging of mice from the ST and DT group was performed once a week beginning at the day of the first uterine tissue transplantation (ST: d0; DT: d-14) (Figure 3C–F). In the ST and DT group, endometriotic lesions developing from uterine tissue samples of transgenic mice exhibited an increased total flux during the first 28 days when compared to d0. Thereafter, the bioluminescence signal markedly decreased to baseline values in the DT group. In contrast, endometriotic lesions developing from uterine tissue samples of wild-type mice showed no signal at the expected locations in both groups during the entire observation period (Figure 3C–F). These findings indicate that there was no relevant exchange of cells from luciferase-positive to luciferase-negative lesions.

### 2.3. Vascularization, Proliferation and Immune Cell Infiltration of Endometriotic Lesions

At the end of our in vivo experiments, the wild-type lesions in the ST and DT group were additionally analyzed by means of immunohistochemistry to assess their vascularization, proliferation and immune cell infiltration. We found that the density of CD31-positive microvessels was comparable in the lesions of both groups (Figure 4A–C). Detection of the proliferation marker Ki67 revealed a significantly higher fraction of proliferating stromal cells inside the lesions of the DT group when compared to those of the ST group (Figure 4D–F). In contrast, no difference could be observed for the glandular cells (Figure 4D–F).

Furthermore, we measured the number of CD3-positive lymphocytes, myeloperoxidase (MPO)-positive neutrophilic granulocytes and CD68-positive macrophages (Figure 5A–G). This analysis showed significantly more lymphocytes infiltrating the wild-type lesions in the DT group when compared to those in the ST group, whereas the number of neutrophilic granulocytes and macrophages was comparable in both groups (Figure 5A–G).

### 2.4. Immune Cells and Inflammatory Factors in the Peritoneal Fluid

To analyze in a final set of experiments the effects of newly developing endometriotic lesions on the physiological microenvironment in the abdominal cavity, a peritoneal lavage was performed in four FVB/N wild-type mice. Thereafter, two uterine tissue samples from a transgenic donor mouse were sutured to the right abdominal wall of each mouse. After a period of 14 days, a peritoneal lavage was performed again. The lavage samples were then further processed for flow cytometry and protein array analysis.

Flow cytometric analyses showed a significantly higher fraction of Gr1-positive granulocytes within the peritoneal fluid 14 days after the induction of endometriotic lesions when compared to baseline conditions (Figure 6A–C). In contrast, no significant differences were observed for the fractions of CD3ε-positive lymphocytes and CD68-positive macrophages (Figure 6D–I).

In addition, we assessed the expression of 40 different cytokines, chemokines and acute phase proteins in the peritoneal fluid by means of a proteome profiler mouse cytokine array. Notably, we found that the expression of most of these factors was increased 14 days after the induction of endometriotic lesions when compared to baseline conditions (Table 1). The strongest upregulation in expression was detected for B lymphocyte chemoattractant (BLC, CXCL13), interleukin (IL)-23, tissue inhibitor of metalloproteinases (TIMP)-1 and IL-5. Among the few downregulated factors, IL-2 and C5/C5a-convertase showed the most pronounced difference in expression when compared to baseline conditions (Table 1).

## 3. Discussion

The pathogenesis of endometriosis is closely associated with repeated retrograde menstruation of endometrial tissue fragments into the peritoneal cavity, where they develop into endometriotic lesions and induce an inflammatory environment [10]. This leads to the assumption that this altered environment may promote the development of new endometriotic lesions besides already established ones and, thus, the progression of the disease in the sense of a vicious circle. In line with this view, we could demonstrate in the present study that the presence of pre-existing endometriotic lesions promotes the growth of new lesions in the peritoneal cavity of mice.

For our proof-of-concept experiments, we used a mouse model of endometriosis in which endometriotic lesions were surgically induced by simultaneous or time-delayed transplantation of uterine tissue samples from donor mice into the peritoneal cavity of recipient animals. Notably, these samples originated from transgenic FVB-Tg(CAG-luc-GFP)L2G85Chco/J mice and FVB/N wild-type mice, which enabled us to analyze the cellular exchange between individual lesions by means of bioluminescence imaging. Moreover, the samples were fixed by sutures in defined locations at the abdominal wall of the recipient animals, which allowed their easy retrieval for high-resolution ultrasound analyses [11]. Both imaging technologies could be performed repeatedly in a non-invasive manner, which markedly reduced the number of animals required for this study in line with the 3R principle. Moreover, high-resolution ultrasound imaging of endometriotic lesions allowed the separate quantification of their stromal and cyst volume, enabling a differentiated statement regarding their tissue proliferation and secretory activity. Our analyses revealed that wild-type endometriotic lesions of the DT group exhibited a significantly higher growth rate when compared to those of the ST group. These results are consistent with our immunohistochemical analyses of the lesions at the end of the in vivo experiments, which showed a higher number of proliferating Ki67-positive cells within the stroma of lesions in the DT group.

To clarify whether the higher growth rate of endometriotic lesions in the DT group was caused by the invasion of endometrial cells from pre-existing luciferase-positive lesions into newly developing luciferase-negative wild-type lesions, we used the technique of bioluminescence imaging, as previously reported in other mouse endometriosis studies [12,13,14]. In our model, we detected an increasing total flux in luciferase-positive lesions of the ST and DT group during the first 28 days after lesion induction. Thereafter, the bioluminescence signal markedly decreased in the DT group. These findings are in line with former studies reporting an increasing bioluminescence signal during the initial phase of lesion development and the progressive loss of signal at later time points [13,14]. Our analyses further showed that endometriotic lesions developing from uterine tissue samples of wild-type mice exhibit no bioluminescence signals during the entire observation period. This indicates that there was no relevant exchange of cells from luciferase-positive to luciferase-negative lesions. Hence, we suggest that the higher growth rate of endometriotic lesions in the DT group must have been caused by other mechanisms, such as an altered inflammatory peritoneal environment. Therefore, we next analyzed inflammatory factors and immune cells in the peritoneal fluid before and after the induction of endometriotic lesions.

The expression of most of the analyzed factors in our proteome profiler mouse cytokine array was increased 14 days after induction of endometriotic lesions when compared to baseline conditions. Some of these factors have previously been shown to be involved in the pathogenesis of endometriosis. For instance, we detected a significant upregulation of IL-5, which is characteristic for the presence of early stage endometriosis and generally contributes to the proliferation, differentiation and survival of different cell types [8,15]. IL-5 is particularly essential for the priming and survival of mature eosinophils, a cell type belonging to the family of granulocytes and their progenitors [16]. This may explain our flow cytometric result that the peritoneal fluid revealed a significantly higher fraction of granulocytes 14 days after lesion induction. Clinical studies could further show that granulocytes are increased in endometriosis patients when compared to healthy women [17,18]. In addition, IL-17, CXCL10 (IP-10) and IL-23 are known to induce inflammation in endometriosis by enhancing the migration of granulocytes [19,20]. Accordingly, we also detected a higher expression of these cytokines in the peritoneal fluid.

We additionally performed immunohistochemical stainings of infiltrated immune cells within wild-type lesions on day 28 after induction. These stainings showed a comparable fraction of granulocytes and macrophages in endometriotic lesions of the ST and DT group. Moreover, we detected significantly more lymphocytes in the lesions of the DT group when compared to those of the ST group. The latter observation is consistent with the elevated IL-23 and IL-17 expression in the peritoneal fluid because activated T-lymphocytes are known to secrete these cytokines [21]. Nonetheless, the overall difference in immune cell infiltration between lesions of the ST and DT group may be too marginal to explain the differences in lesion growth between the two groups. Therefore, we suggest that the higher proliferation rate of stromal cells within the lesions of mice in the DT group is rather caused by a direct stimulatory effect of the cytokines in the peritoneal fluid of these animals. In line with this view, previous studies already reported that cytokines, such as IL-1β and IL-4, promote the proliferating activity of isolated endometrial stroma cells [22,23].

Finally, it should be mentioned that the present study has also some limitations. In our model, we surgically induced endometriotic lesions by transplanting uterine tissue samples of healthy donor mice without the use of pathological endometriotic tissue of human origin. Therefore, the results obtained in this mouse model may not fully correlate to human patients with endometriosis. Moreover, due to ethical reasons the number of analyzed mice was reduced to a minimum, which may limit the statistical significance of our results. Furthermore, we analyzed the growth of newly developing endometriotic lesions only within 28 days. Hence, we cannot exclude the fact that longer observation time periods would have shown different results.

## 4. Materials and Methods

### 4.1. Animals

For this study, 12- to 20-week-old female FVB/N wild-type mice and FVB-Tg(CAG-luc-GFP)L2G85Chco/J transgenic mice (Institute for Clinical and Experimental Surgery, Homburg, Germany) with a body weight of 18–25 g were used. The animals were housed in groups on wooden chip bedding and had free access to standard pellet food (Altromin, Lage, Germany) and water. They were maintained in the conventional animal facility of the Institute for Clinical and Experimental Surgery under a 12 h day-night cycle.

### 4.2. Vaginal Lavage

To ensure similar sex hormone levels in individual mice, estrous cycle stages were cytologically assessed by analyzing vaginal lavage samples. For this purpose, 15 µL of 0.9% saline were carefully pipetted into the vagina and transferred to a glass slide for examination under a phase-contrast microscope (CH-2; Olympus, Hamburg, Germany) [24]. Only those mice, which were in the estrus stage, were selected as donors for uterine tissue samples as well as recipient animals for the induction of endometriotic lesions.

### 4.3. Isolation of Uterine Tissue Samples

Uterine tissue samples for the induction of peritoneal endometriotic lesions were excised from FVB/N wild-type and transgenic FVB-Tg(CAG-luc-GFP)L2G85Chco/J mice in the stage of estrus. The mice were anesthetized by an intraperitoneal injection of 100 mg/kg ketamine (Ursotamin^®^; Serumwerke Bernburg, Bernburg, Germany) and 12 mg/kg xylazine (Rompun^®^; Bayer, Leverkusen, Germany). Uterine horns were isolated after midline laparotomy, placed into a Petri dish containing Dulbecco’s modified Eagle medium (DMEM; PAN Biotech, Aidenbach, Germany; 10% fetal calf serum, 100 U/mL penicillin, 0.1 mg/mL streptomycin (Thermo Fisher Scientific, Dreieich, Germany)) and opened longitudinally. Subsequently, 2 mm tissue samples were carefully removed by means of a dermal biopsy punch (Stiefel Laboratorium GmbH, Offenbach am Main, Germany).

### 4.4. Induction of Peritoneal Endometriotic Lesions

For the surgical induction of endometriotic lesions, uterine tissue samples from transgenic and wild-type mice were transplanted into the abdominal cavity of wild-type mice, as previously described [25]. The mice were anesthetized by an intraperitoneal injection of 100 mg/kg ketamine (Ursotamin^®^, Serumwerke Bernburg) and 12 mg/kg xylazine (Rompun^®^, Bayer). After midline laparotomy, the uterine tissue samples were fixed with 6-0 Seramon sutures (Serag-Wiessner Products, Naila, Germany) to the abdominal wall. Thereafter, the laparotomy was closed again with running Prolene sutures (Ethicon Products, Norderstedt, Germany).

### 4.5. High-Resolution Ultrasound Imaging

By means of a Vevo LAZR system (FUJIFILM VisualSonics Inc.; Toronto, ON, Canada) equipped with a 256-element linear-array transducer (LZ 550, FUJIFILM VisualSonics Inc.) and set on a center frequency of 40 MHz, the development of surgically induced endometriotic lesions was repeatedly analyzed. For this purpose, the mice were anesthetized with 2% isoflurane in oxygen, fixed in supine position on a heated stage and the abdomen was chemically depilated (Veet Hair Removal Cream; Reckitt Benckiser Germany, Heidelberg, Germany). Using the Vevo imaging station (FUJIFILM VisualSonics Inc.; Toronto, ON, Canada) with a mouse platform, the heart and respiratory rate was monitored while maintaining a body temperature of 36–37 °C. After applying the ultrasound gel, the scan head was linearly shifted by a 3D-motor over the abdomen and two-dimensional parallel images of the endometriotic lesions were acquired in intervals of 50 µm.

To analyze the ultrasound images, a three-dimensional reconstruction and analysis software was used (version VevoLAB 5.6.1; VisualSonics). The borders of the lesions and their cyst-like dilated endometrial glands were outlined in parallel sections with a step size of 200 µm and the total volume of the developing endometriotic lesions as well as the volume of their stromal tissue and cysts (in mm^3^) were calculated by manual image segmentation [26]. Furthermore, the growth rate of the lesions and stromal tissue (in % of the initial lesion and stromal tissue volume) as well as the fraction of cyst-containing lesions (in %) were calculated. At the end of the in vivo experiments, the anesthetized animals were carefully laparotomized under a stereo-microscope. Subsequently, the lesions were harvested and fixed in paraformaldehyde for additional histological and immunohistochemical analyses.

### 4.6. In Vivo and Ex Vivo Bioluminescence Imaging

The visualization of luciferase activity was performed by means of bioluminescence imaging with an IVIS Spectrum In Vivo Imaging System (IVIS Spectrum, Perkin Elmer, Waltham, MA, USA). After anesthesia with 2% isoflurane in oxygen, mice were injected subcutaneously with 50 mg/kg synthetic D-luciferin (CycLuc1, Tocris, Bio-Techne; Wiesbaden, Germany) in 100 µL phosphate-buffered saline (PBS). After an incubation period of 5 min to ensure the vascular distribution of the substrate, the animals were placed inside the IVIS Spectrum and imaged using the Living Image software (version 4.7.3; Perkin Elmer). The bioluminescence of each lesion was assessed by drawing a ROI (region of interest) over the lesion location inside the abdomen. Within the chosen ROIs, the total flux was measured in photons/second (p/s).

### 4.7. Histology and Immunohistochemistry

The paraformaldehyde-fixed specimens of endometriotic lesions were embedded in paraffin. Three-µm-thick sections were cut and stained with HE according to standard procedures.

For the immunohistochemical detection of proliferating cells, as well as lymphocytes, neutrophilic granulocytes and macrophages within the lesions, sections were stained with a rabbit polyclonal antibody against the proliferation marker Ki67 (1:400; Cell Signaling, Leiden, The Netherlands), a rabbit polyclonal antibody against the lymphocyte marker CD3 (1:100; Abcam, Cambridge, UK), a rabbit polyclonal antibody against the neutrophilic granulocyte marker MPO (1:100; Abcam) and a rabbit polyclonal antibody against the macrophage marker CD68 (1:300; Abcam). As secondary antibodies, a goat anti-rabbit biotinylated antibody (ready-to-use, Abcam) followed by avidin-peroxidase (ready-to-use, Abcam) was used. Then, 3-amino-9-ethylcarbazole (AEC Substrate System, Bio SB Inc., Santa Barbara, CA, USA) served as chromogen followed by a counterstaining with hemalaun. The fraction of proliferating cells (%) was assessed by counting the number of Ki67-positive stromal and glandular cells in four regions of interest within the endometriotic lesions. The number of CD3-positive lymphocytes, MPO-positive neutrophilic granulocytes and CD68-positive macrophages (mm^−2^) was assessed by counting the positive cells in four ROIs within the endometriotic lesions.

Moreover, microvessels were detected by staining sections with a monoclonal rat anti-mouse antibody against the endothelial cell marker CD31 (1:100; Dianova GmbH, Hamburg, Germany). A goat anti-rat IgG Alexa555 antibody (1:100; Invitrogen, Darmstadt, Germany) served as a secondary antibody. Cell nuclei were stained with Hoechst 33342 (2 µg/mL; Sigma-Aldrich). The microvessel density (mm^−2^) was measured using a BZ-8000 microscope (Keyence, Osaka, Japan). For this purpose, the overall number of CD31-positive microvessels was counted and divided by the analyzed area of stromal tissue.

### 4.8. Peritoneal Lavage

To analyze the peritoneal milieu before and after the induction of endometriotic lesions, a peritoneal lavage was performed. For this purpose, mice were anesthetized by intraperitoneal injection of 100 mg/kg ketamine (Ursotamin^®^; Serumwerke Bernburg, Germany) and 12 mg/kg xylazine (Rompun^®^; Bayer). Subsequently, 2 mL of ice-cold PBS were injected into the peritoneal cavity and distributed by gentle massage of the abdomen to dislocate any attached cells. Two minutes later, the cell suspension was extracted with a 20G cannula (B. Braun; Melsungen, Germany) and stored on ice.

### 4.9. Flow Cytometry

The collected peritoneal fluid was centrifuged for 5 min at 400× *g* to separate the supernatant from the cellular precipitate. For flow cytometry, the cell suspension was incubated for 30 min with fluorescein isothiocyanate (FITC)-conjugated monoclonal rat anti-mouse granulocyte surface marker Gr-1 (ImmunoTools; Friesoythe, Germany) and FITC-conjugated monoclonal hamster anti-mouse lymphocyte surface marker CD3ε (ImmunoTools). Subsequently, cells were fixed and permeabilized using BD Cytofix/Cytoperm™ Fixation and Permeabilization Solution (BD Biosciences; Heidelberg, Germany) and incubated with the PE-conjugated monoclonal rat anti-mouse intracellular macrophage marker CD68 (BD Pharmingen, BD Biosciences). Flow cytometric analysis was performed by means of a FACSLyric system (BD Biosciences) and data were assessed with the FACSuite software package (version 1.3; BD Biosciences).

### 4.10. Cytokine Array

The supernatant obtained after centrifugation of the peritoneal fluid was analyzed by means of a membrane-based sandwich immunoassay using a proteome profiler mouse cytokine array kit (R&D Systems, Bio-Techne; Wiesbaden, Germany). Samples were mixed with a cocktail of 40 mouse cytokines, chemokines and acute phase proteins and incubated with the array membrane according to manufacturer’s instructions. The proteins were made visible by means of chemiluminescent detection reagents.

### 4.11. Statistics

All experiments were designed to generate groups of equal size, using randomization and blinded analysis. Data were first analyzed for normal distribution and equal variance. In the case of parametric data, differences between groups were assessed by an unpaired Student’s *t*-test. In case of non-parametric data, differences between groups were assessed by a Mann–Whitney rank sum test. To test for time effects within each experimental group, ANOVA for repeated measurements was applied for parametric data followed by Tukey’s post hoc test. Non-parametric data were analyzed by Friedman’s test followed by Dunn’s post hoc test (GraphPad Prism 9.5.1; GraphPad Software, San Diego, CA, USA). All data are given as mean ± standard error of the mean (SEM). Statistical significance was accepted for *p* < 0.05.

## 5. Conclusions

Our study demonstrates that pre-existing endometriotic lesions promote the growth of new lesions in the peritoneal cavity of mice. This is not associated with a relevant cellular exchange between the two lesion types but rather mediated by peritoneal inflammation, which creates favorable conditions for the development of new lesions. Translated to the clinical situation, these findings indicate that patients already suffering from peritoneal endometriosis may have a higher risk for the establishment and progression of additional endometriotic lesions in the sense of a vicious circle. Considering the fact that the diagnosis of endometriosis is often delayed for many years due to its unspecific symptoms [27,28], this view underlines the importance of developing faster diagnostic routines for the disease which enable the rapid eradication of endometriotic lesions within the peritoneal cavity. For this purpose, the identification of disease-specific molecular diagnostic markers within the eutopic endometrium of endometriosis patients or their plasma may be a promising approach [29,30]. Moreover, such markers may also help to improve the risk stratification and therapeutic management of endometriosis as it is already established for the management of endometrial cancer [31].

## Figures and Tables

**Figure 1 ijms-24-13858-f001:**
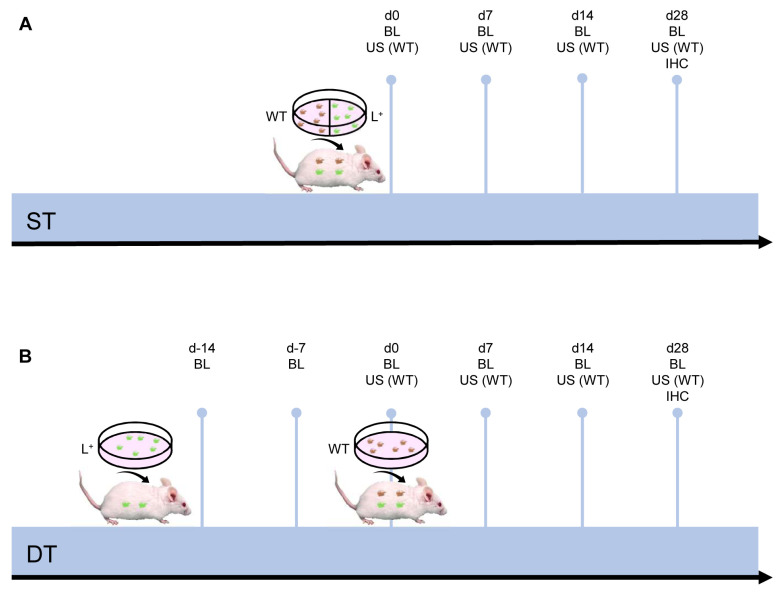
Schematic illustration of the study design. (**A**) Simultaneous transplantation (ST) of uterine tissue samples from FVB-Tg(CAG-luc-GFP)L2G85Chco/J (L^+^) and FVB/N wild-type (WT) mice into the peritoneal cavity of FVB/N wild-type mice. (**B**) Time-delayed transplantation (DT) of uterine tissue samples from FVB-Tg(CAG-luc-GFP)L2G85Chco/J on d-14 and FVB/N wild-type mice on d0 into the peritoneal cavity of FVB/N wild-type mice. All lesions were analyzed by bioluminescence (BL). Wild-type lesions were additionally imaged by high-resolution ultrasound (US). At the end of the in vivo experiments, all lesions were excised and further processed for histology and immunohistochemistry (IHC).

**Figure 2 ijms-24-13858-f002:**
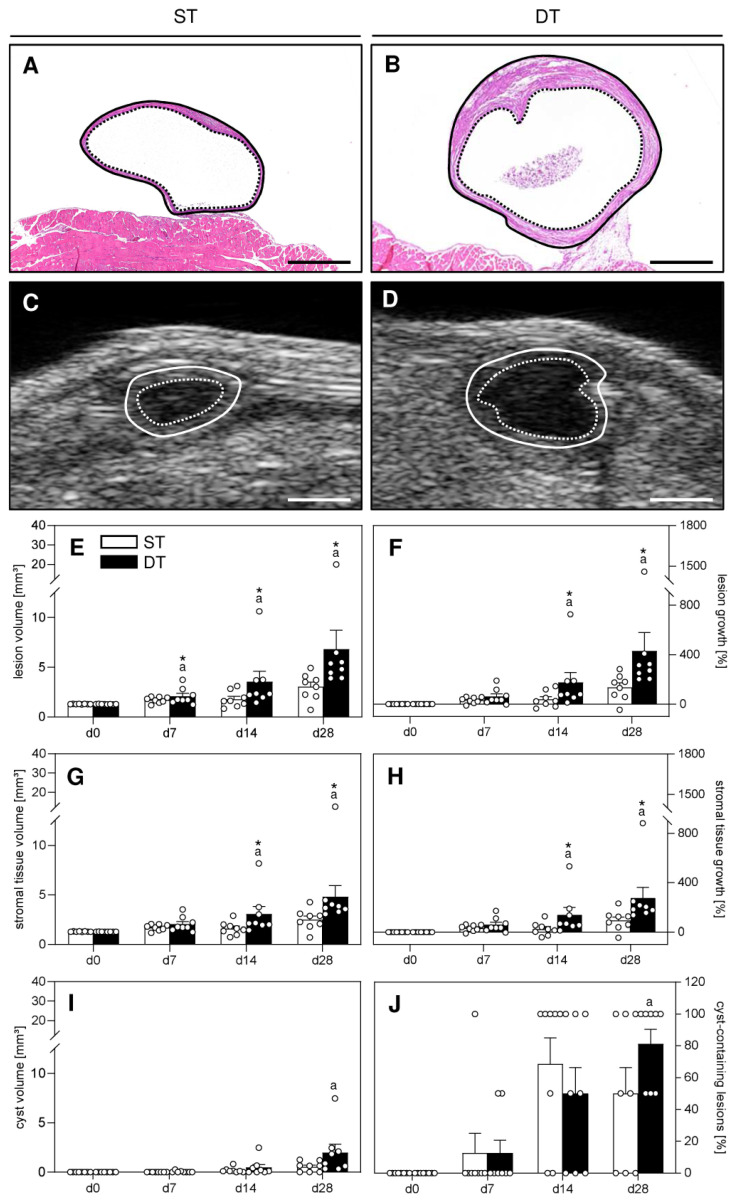
Histomorphology and high-resolution ultrasound imaging of endometriotic lesions. (**A**,**B**) HE-stained sections of wild-type endometriotic lesions (borders marked by lines, cyst-like dilated endometrial glands marked by dotted lines) 28 days after simultaneous (ST; (**A**)) and time-delayed (DT; (**B**)) transplantation of uterine tissue samples from FVB/N wild-type mice into the peritoneal cavity of FVB/N wild-type mice. Scale bars: (**A**,**B**) = 500 µm. (**C**,**D**) High-resolution ultrasound imaging of wild-type endometriotic lesions (borders marked by lines, cyst-like dilated endometrial glands marked by dotted lines) 28 days after simultaneous (ST; (**C**)) and delayed (DT; (**D**)) transplantation of uterine tissue samples from FVB/N wild-type mice into the peritoneal cavity of FVB/N wild-type mice. Scale bars: (**C**,**D**) = 1 mm. (**E**–**J**) Lesion volume ((**E**), mm^3^), lesion growth ((**F**), %), stromal tissue volume ((**G**), mm^3^), stromal tissue growth (**H**), %), cyst volume ((**I**), mm^3^) and fraction of cyst-containing lesions ((**J**), %) of wild-type endometriotic lesions induced by transplantation of uterine tissue samples from FVB/N wild-type mice into the peritoneal cavity of FVB/N wild-type mice of the ST group (white bars; n = 8) and DT group (black bars; n = 8). Mean ± SEM; ^a^ *p* < 0.05 vs. d0; * *p* < 0.05 vs. ST.

**Figure 3 ijms-24-13858-f003:**
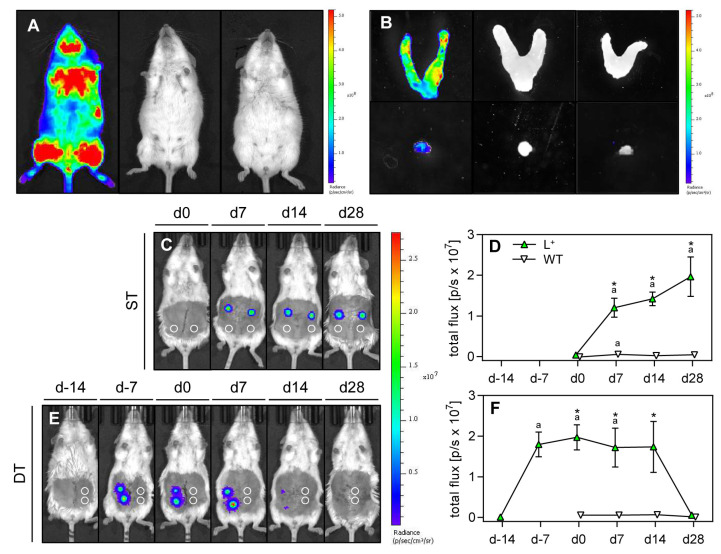
Bioluminescence imaging of endometriotic lesions. (**A**) Left to right: FVB-Tg(CAG-luc-GFP)L2G85Chco/J mouse after injection of synthetic D-luciferin; FVB-Tg(CAG-luc-GFP)L2G85Chco/J mouse without injection of synthetic D-luciferin; FVB/N wild-type mouse after injection of synthetic D-luciferin. (**B**) Left to right: Uteri (upper panel) and uterine tissue samples (lower panel) of mice in A. (**C**) Whole body bioluminescence images of FVB/N wild-type mice 28 days after simultaneous transplantation (ST) of uterine tissue samples from FVB-Tg(CAG-luc-GFP)L2G85Chco/J mice and FVB/N wild-type mice (location marked by circles) into the peritoneal cavity of FVB/N wild-type mice. (**D**) Total flux (p/s × 10^7^) of bioluminescence in uterine tissue samples from FVB-Tg(CAG-luc-GFP)L2G85Chco/J mice (L^+^, green triangles; n = 8) and FVB/N wild-type mice (WT, white triangles; n = 8) after simultaneous transplantation into the peritoneal cavity of FVB/N wild-type mice. ^a^
*p* < 0.05 vs. d0; * *p* < 0.05 vs. WT. (**E**) Whole body bioluminescence images of FVB/N wild-type mice 28 days after time-delayed transplantation (DT) of uterine tissue samples from FVB-Tg(CAG-luc-GFP)L2G85Chco/J mice (d-14) and FVB/N wild-type mice (location marked by circles) (d0) into the peritoneal cavity of FVB/N wild-type mice. (**F**) Total flux (p/s × 10^7^) of bioluminescence in uterine tissue samples from FVB-Tg(CAG-luc-GFP)L2G85Chco/J mice (green triangles; n = 8) and FVB/N wild-type mice (white triangles; n = 8) after time-delayed transplantation into the peritoneal cavity of FVB/N wild-type mice. ^a^ *p* < 0.05 vs. d-14; * *p* < 0.05 vs. WT.

**Figure 4 ijms-24-13858-f004:**
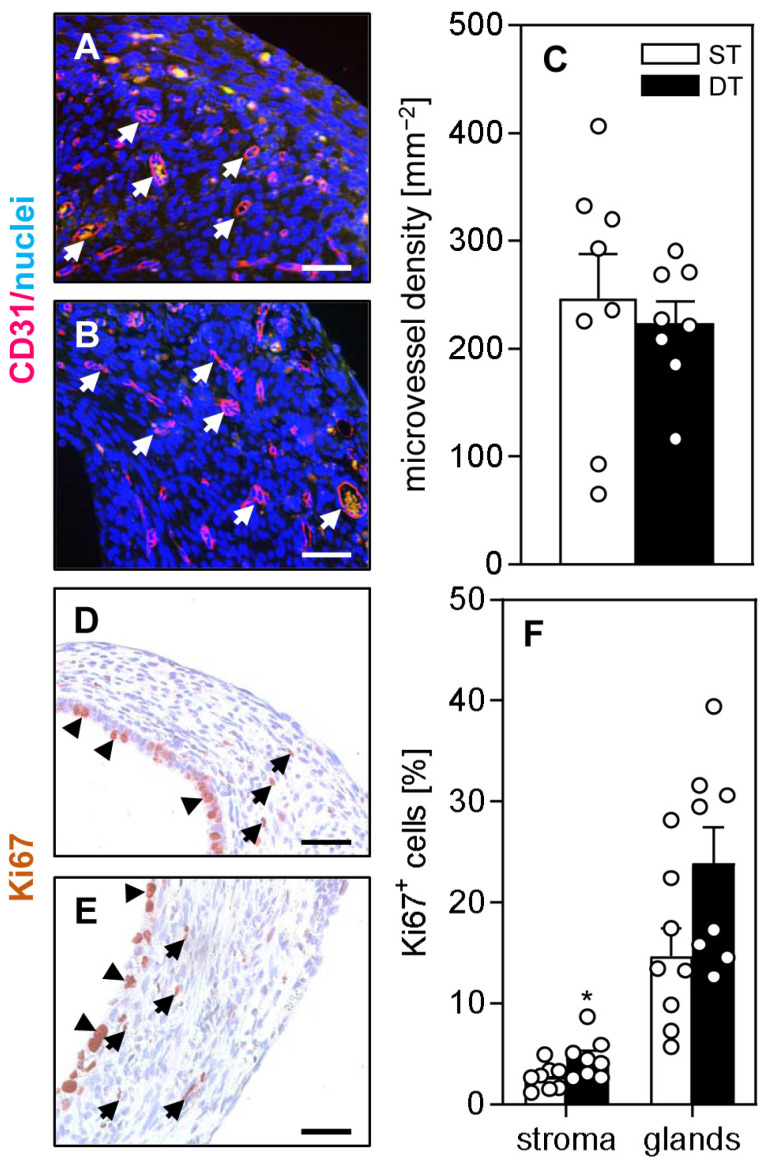
Immunohistochemical analysis of vascularization and proliferation of endometriotic lesions. (**A**,**B**) Immunofluorescent detection of microvessels (arrows) in wild-type endometriotic lesions on day 28 after simultaneous (ST; (**A**)) and time-delayed (DT; (**B**)) transplantation of uterine tissue samples from FVB-Tg(CAG-luc-GFP)L2G85Chco/J mice and FVB/N wild-type mice into the peritoneal cavity of FVB/N wild-type mice. The immunofluorescent sections were stained with Hoechst 33342 to identify cell nuclei (blue) and an antibody against CD31 for the detection of microvessels (red). Scale bars: 50 µm. (**C**) Microvessel density (mm^−2^) of endometriotic lesions induced by transplantation of uterine tissue samples from FVB/N wild-type mice into the peritoneal cavity of FVB/N wild-type mice in the ST group (white bars; n = 8) and DT group (black bars; n = 8). Mean ± SEM. (**D**,**E**) Immunohistochemical detection of Ki67-positive stromal cells (arrows) and glandular epithelial cells (arrowheads) in wild-type endometriotic lesions on day 28 after simultaneous (ST; (**D**)) and time-delayed (DT; (**E**)) transplantation of uterine tissue samples from FVB-Tg(CAG-luc-GFP)L2G85Chco/J mice and FVB/N wild-type mice into the peritoneal cavity of FVB/N wild-type mice. The immunofluorescent sections were stained with an antibody against Ki67 for the detection of proliferating cells. Scale bars: 50 µm. (**F**) Ki67-positive cells (%) within the stroma and the glands of endometriotic lesions induced by transplantation of uterine tissue samples from FVB/N wild-type mice into the peritoneal cavity of FVB/N wild-type mice in the ST group (white bars; n = 8) and DT group (black bars; n = 8). Mean ± SEM. * *p* < 0.05 vs. ST.

**Figure 5 ijms-24-13858-f005:**
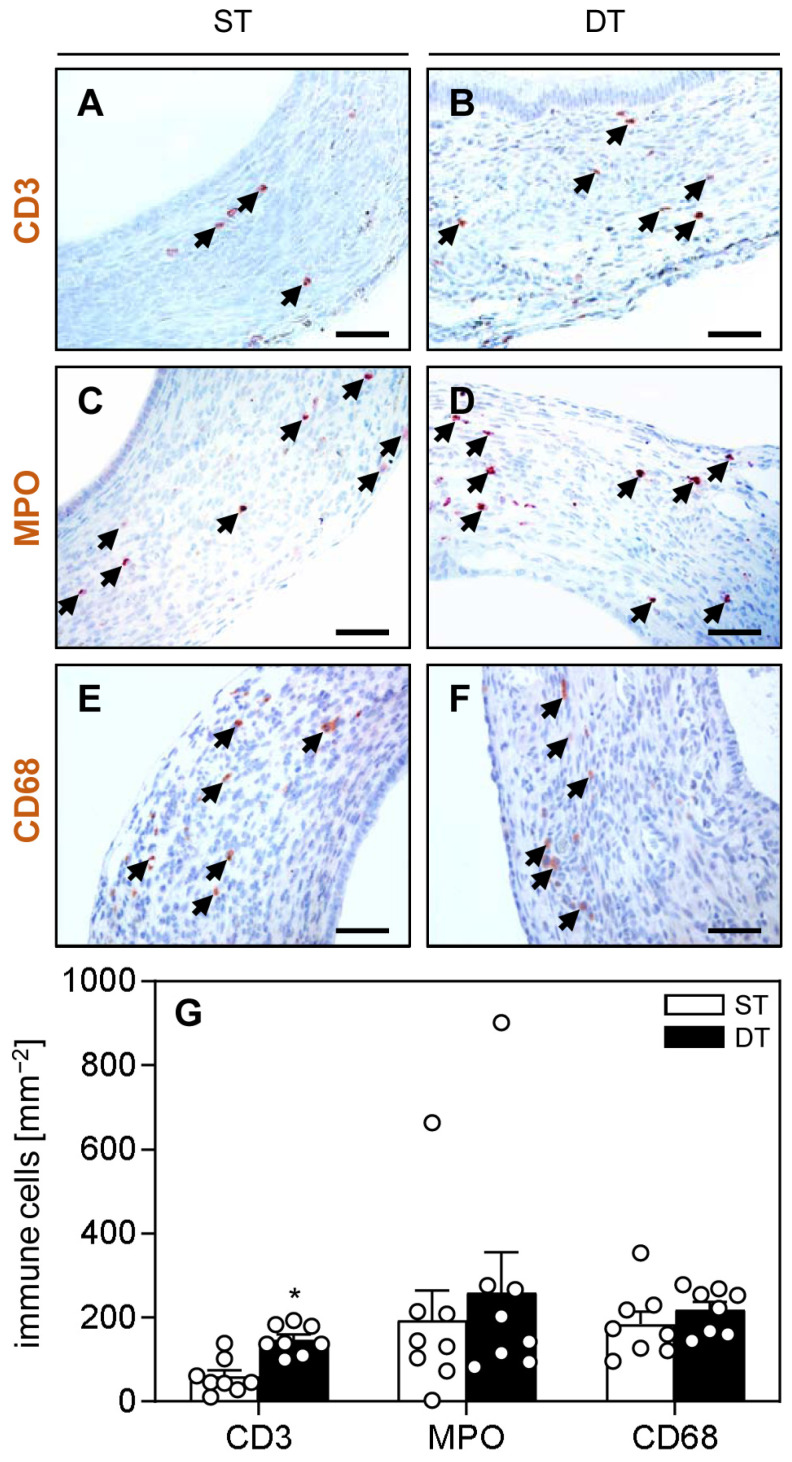
Immunohistochemical analysis of immune cell infiltration into endometriotic lesions. (**A**–**F**) Immunohistochemical detection of CD3positive lymphocytes ((**A**,**B**); arrows), MPO-positive neutrophilic granulocytes ((**C**,**D**); arrows) and CD68-positive macrophages ((**E**,**F**); arrows) in wild-type endometriotic lesions on day 28 after simultaneous (ST; (**A**,**C**,**E**)) and time-delayed (DT; (**B**,**D**,**F**)) transplantation of uterine tissue samples from FVB-Tg(CAG-luc-GFP)L2G85Chco/J mice and FVB/N wild-type mice into the peritoneal cavity of FVB/N wild-type mice. Scale bars: 50 µm. (**G**) CD3-positive lymphocytes (mm^−2^), MPO-positive neutrophilic granulocytes (mm^−2^) and CD68-positive macrophages (mm^−2^) in endometriotic lesions induced by transplantation of uterine tissue samples from FVB/N wild-type mice into the peritoneal cavity of FVB/N wild-type mice in the ST group (white bars; n = 8) and DT group (black bars; n = 8). Mean ± SEM; * *p* < 0.05 vs. ST.

**Figure 6 ijms-24-13858-f006:**
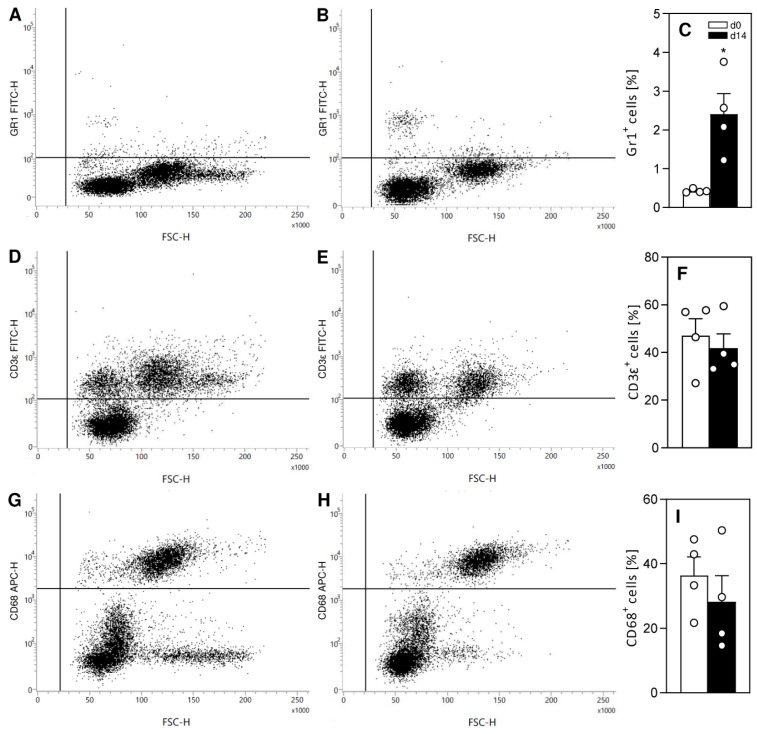
Flow cytometric analysis of peritoneal fluid. (**A**,**B**,**D**,**E**,**G**,**H**) Representative scatter plots of flow cytometric analyses of the peritoneal fluid before (d0; (**A**,**D**,**G**)) and 14 days after (d14; (**B**,**E**,**H**)) transplantation of uterine tissue samples from FVB-Tg(CAG-luc-GFP)L2G85Chco/J mice into the peritoneal cavity of FVB/N wild-type mice. Cells were stained with markers against Gr1 (granulocytes; (**A**,**B**)), CD3ε (lymphocytes; (**D**,**E**)) and CD68 (macrophages; (**G**,**H**)). (**C**,**F**,**I**) Fraction (%) of Gr1-positive granulocytes (**C**), CD3ε-positive lymphocytes (**F**) and CD68-positive macrophages (**I**) in the peritoneal fluid of FVB/N wild-type mice before (d0; white bars; n = 4) and 14 days after (d14; black bars; n = 4) transplantation of uterine tissue samples from FVB-Tg(CAG-luc-GFP)L2G85Chco/J mice into the peritoneal cavity of FVB/N wild-type mice. Mean ± SEM; * *p* < 0.05 vs. d0.

**Table 1 ijms-24-13858-t001:** Expression of different factors in the peritoneal fluid before (d0) and 14 days (d14) after induction of endometriotic lesions, as assessed by a proteome profiler mouse cytokine array. Data are presented as mean pixel density ± SEM (n = 4) of two technical replicates and as ratio d14/d0 in %. * *p* < 0.05 vs. d0.

Protein	Mean Pixel Density (Mean ± SEM)	Mean Pixel Density (Mean ± SEM)	Ratio d14/d0
	d0	d14	(%)
BLC (CXCL13)	3139 ± 457	11,347 ± 5196	361
IL-23	2608 ± 387	5955 ± 799	228
TIMP-1	17,609 ± 6297	38,341 ± 8289	217 *
IL-1b	2742 ± 414	5809 ± 1851	211
IL-5	2054 ± 607	4313 ± 575	210 *
MCP-5	2515 ± 943	4970 ± 800	197
IL-16	5427 ±1826	10,451 ± 5466	192
IL-12p70	2475 ± 652	4381 ± 922	177
I-309	2868 ± 659	4896 ± 991	170
IL-17	3583 ± 988	6103 ± 1791	170
IP-10	3483 ± 1227	5395 ± 1361	154
IL-4	2545 ± 683	3873 ± 492	152
IL-3	3107 ± 579	4304 ± 1164	138
IFN-gamma	9087 ± 1316	12,272 ± 3164	135
ICAM-1	24,038 ± 8847	31,685 ± 8232	131
G-CSF	4973 ± 784	6291 ± 2001	126
IL-1a	4120 ± 514	5041 ± 1157	122
IL-10	3148 ± 1159	3836 ± 1130	121
IL-7	3724 ± 857	4445 ± 1501	119
MIP-1a	3084 ± 549	3533 ± 835	114
TARC	3200 ± 1242	3481 ± 930	108
JE	4832 ± 2132	5195 ± 833	107
IL-6	2701 ± 1058	2894 ± 518	107
M-CSF	4425 ± 1579	4660 ± 1964	105
Eotaxin	4012 ± 789	4215 ± 804	105
TREM-1	2833 ± 325	2956 ± 789	104
RANTES	4670 ± 1873	4866 ± 1422	104
MIP-2	3553 ± 718	3610 ± 901	101
MIG	4083 ± 973	4129 ± 905	101
SDF-1	3447 ± 348	3480 ± 750	100
IL-27	4651 ± 1237	4307 ± 1550	92
MIP-1b	3268 ± 722	2887 ± 447	88
IL-13	3276 ± 814	2672 ± 690	81
GM-CSF	3066 ± 766	2416 ± 526	78
I-TAC	5839 ± 1395	4521 ± 962	63
TNF-a	3478 ± 1067	2629 ± 1019	59
KC	3109 ± 742	2310 ± 760	54
IL-1ra	10,822 ± 6491	5344 ± 1213	50
C5/C5a	6910 ± 516	2819 ± 607	46 *
IL-2	6513 ± 1015	2259 ± 562	42 *

## Data Availability

The data that support the findings of this study are available from the corresponding author upon reasonable request.

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
