# Peer review of "The Presence of Pre-Existing Endometriotic Lesions Promotes the Growth of New Lesions in the Peritoneal Cavity"

_ijms, 2023, doi:10.3390/ijms241813858_

Round 1

Reviewer 1 Report

Mihai et al described the relationship between preexisting endometriosis lesions and the growth of the new lesion in the peritonea cavities.

The endometriosis is a public health problem that has an impact not only on women's health but also important socio-economic repercussions. This article contributes to the advancement of our knowledge of this disease. The article is well organized and it will be an update.

However, the implication of these results in the management and stratification of patients with peritoneal lesions is lacking. The authors should discuss these results in terms of the prevalence of peritoneal lesions in women after primary endometriosis.

The conclusion is missing

the article is well written 

Author Response

Review of the manuscript ID ijms-2551708 by Mihai et al.

Reply to the comments of reviewer 1

We appreciate the fair and constructive comments of the reviewer. In the following, please find our point-by-point reply.

1. Reviewer comment: Mihai et al described the relationship between preexisting endometriosis lesions and the growth of the new lesion in the peritonea cavities.

The endometriosis is a public health problem that has an impact not only on women's health but also important socio-economic repercussions. This article contributes to the advancement of our knowledge of this disease. The article is well organized and it will be an update.

However, the implication of these results in the management and stratification of patients with peritoneal lesions is lacking. The authors should discuss these results in terms of the prevalence of peritoneal lesions in women after primary endometriosis.

Reply: According to the comment of the reviewer, we have included an additional paragraph in the new conclusion section of our manuscript, which discusses our results in terms of the prevalence of peritoneal lesions in women after primary endometriosis. This paragraph reads as follows:

‘Translated to the clinical situation these findings indicate that patients already suffering from peritoneal endometriosis may have a higher risk for the establishment and progression of additional endometriotic lesions in the sense of a vicious circle. Considering the fact that the diagnosis of endometriosis is often delayed for many years due to its unspecific symptoms (1,2), this view underlines the importance to develop faster diagnostic routines for the disease, which enable a rapid eradication of endometriotic lesions within the peritoneal cavity.’

(See lines 345-352; lines 583-586; marked in yellow)

References:

  1. Davenport, S.; Smith, D.; Green, D.J. Barriers to a Timely Diagnosis of Endometriosis: A Qualitative Systematic Review. Obstet Gynecol. 2023; 142,571-583.
  2. Moein Mahini, S.; Younesi, M.; Mortazavi, G.; Samare-Najaf, M.; Karim Azadbakht, M.; Jamali, N. Non-invasive diagnosis of endometriosis: Immunologic and genetic markers. Clin Chim Acta. 2023; 538,70-86.

2. Reviewer comment: The conclusion is missing.

Reply: According to the comment of the reviewer, we have included a conclusion in the revised version of our manuscript (see lines 341-356; marked in yellow).

Reviewer 2 Report

‘’ The presence of pre-existing endometriotic lesions promotes the growth of new lesions in the peritoneal cavity’’

The manuscript presented for review consists of 16 pages with 26 references. 1 table and 6 figures are included. The study is original. Each of the figures has been prepared and designed with precision. They properly show obtained data. The manuscript is divided into 4 sections. 

- I would suggest to change the order of chapters : Introduction, Material and Methods, Results, Discussion, to present how the study was designed, conducted and what are the conclusions.

Abstract is covered main contents. Background is good arranged.

- I would recommend to expand literature review.

Methods are routine and logical. Statistical tests are correct. Results show findings clearly. Conclusion is appropriate.

This study have potential to improve the knowledge concerning etiology and pathogenesis of endometriosis. I appreciate the contribution and commitment of all authors. In my opinion this manuscript is well - organized and it could be accepted after minor revisions.

Author Response

Review of the manuscript ID ijms-2551708 by Mihai et al.

Reply to the comments of reviewer 2

We appreciate the fair and constructive comments of the reviewer. In the following, please find our point-by-point reply.

1. Reviewer comment: The manuscript presented for review consists of 16 pages with 26 references. 1 table and 6 figures are included. The study is original. Each of the figures has been prepared and designed with precision. They properly show obtained data. The manuscript is divided into 4 sections.

I would suggest to change the order of chapters: Introduction, Material and Methods, Results, Discussion, to present how the study was designed, conducted and what are the conclusions.

Reply: We have arranged the chapters of our manuscript exactly according to the IJMS Microsoft Word template file that is provided by the journal in the ‘Instructions for Authors’. In this file the order of the chapters is clearly given as Introduction, Results, Discussion, Conclusion, Materials and Methods. Therefore, we have decided not to change the order of the different chapters.  

2. Reviewer comment: I would recommend to expand literature review.

Reply: According to the comment of the reviewer, we have included additional literature in the revised version of our manuscript (see lines 583-594; marked in yellow).  

Reviewer 3 Report

I read with great interest the manuscript, which falls within the aim of this Journal and offers a high-quality overview of the topic.  

Although the manuscript can be considered already of high quality, I would suggest taking into account the following minor recommendations:

- I suggest another language revision round to correct a few typos and improve readability.

- I invite authors to read authors' guidelines to better organize their manuscript.

- I find it interesting to include a reference to the molecular mechanism involved in endometrial pathology that can help their diagnosis and role treatments (see PMID: 36979434 and PMID: 36833105).

- What are the actual clinical implications of this study? it is important to report the results obtained by the authors in the context of clinical practice and to adequately highlight what contribution this study adds to the literature already existing on the topic and to future study perspectives.

- The authors have not adequately highlighted the strengths and limitations of their study. I suggest better specifying these points.

The whole text should be corrected by a native English speaker in order to make the work clearer and more readable.

Author Response

Review of the manuscript ID ijms-2551708 by Mihai et al.

Reply to the comments of reviewer 3

We appreciate the fair and constructive comments of the reviewer. In the following, please find our point-by-point reply.

1. Reviewer comment: I suggest another language revision round to correct a few typos and improve readability.

Reply: According to the comment of the reviewer, we have again checked the manuscript for correct English language and typos.   

2. Reviewer comment: I invite authors to read authors' guidelines to better organize their manuscript.

Reply: We have arranged the chapters of our manuscript exactly according to the IJMS Microsoft Word template file that is provided by the journal in the ‘Instructions for Authors’.

3. Reviewer comment: I find it interesting to include a reference to the molecular mechanism involved in endometrial pathology that can help their diagnosis and role treatments (see PMID: 36979434 and PMID: 36833105).

Reply: According to the comment of the reviewer, we have included a novel paragraph with additional references in the revised version of our manuscript, which discusses the relevance of molecular mechanisms for the future diagnosis and therapy of endometriosis. This paragraph reads as follows:

‘For this purpose, the identification of disease-specific molecular diagnostic markers within the eutopic endometrium of endometriosis patients or their plasma may be a promising approach (1,2). Moreover, such markers may also help to improve the risk stratification and therapeutic management of endometriosis, as it is already established for the management of endometrial cancer (3).’

(See lines 352-356; lines 587-594; marked in yellow)

References:

  1. Holzer, I.; Machado Weber, A.; Marshall, A.; Freis, A.; Jauckus, J.; Strowitzki, T.; Germeyer, A. GRN, NOTCH3, FN1, and PINK1 expression in eutopic endometrium - potential biomarkers in the detection of endometriosis - a pilot study. J Assist Reprod Genet. 2020; 37,2723-2732.
  2. Zubrzycka, A.; Migdalska-Sęk, M.; Jędrzejczyk, S.; Brzeziańska-Lasota, E. Circulating miRNAs Related to Epithelial-Mesenchymal Transitions (EMT) as the New Molecular Markers in Endometriosis. Curr Issues Mol Biol. 2021; 43,900-916.
  3. Cuccu, I.; D'Oria, O.; Sgamba, L.; De Angelis, E.; Golia D'Augè, T.; Turetta, C.; Di Dio, C.; Scudo, M.; Bogani, G.; Di Donato, V.; Palaia, I.; Perniola, G.; Tomao, F.; Muzii, L.; Giannini, A. Role of Genomic and Molecular Biology in the Modulation of the Treatment of Endometrial Cancer: Narrative Review and Perspectives. Healthcare (Basel). 2023; 11,571.

4. Reviewer comment: What are the actual clinical implications of this study? it is important to report the results obtained by the authors in the context of clinical practice and to adequately highlight what contribution this study adds to the literature already existing on the topic and to future study perspectives.

Reply: According to the comment of the reviewer and another comment of reviewer 1, we have included an additional paragraph in the new conclusion section of our manuscript, which discusses our results in the context of clinical practice. This paragraph reads as follows:

‘Translated to the clinical situation these findings indicate that patients already suffering from peritoneal endometriosis may have a higher risk for the establishment and progression of additional endometriotic lesions in the sense of a vicious circle. Considering the fact that the diagnosis of endometriosis is often delayed for many years due to its unspecific symptoms (1,2), this view underlines the importance to develop faster diagnostic routines for the disease, which enable a rapid eradication of endometriotic lesions within the peritoneal cavity.’

(See lines 345-352; lines 383-386; marked in yellow)

References:

  1. Davenport, S.; Smith, D.; Green, D.J. Barriers to a Timely Diagnosis of Endometriosis: A Qualitative Systematic Review. Obstet Gynecol. 2023; 142,571-583.
  2. Moein Mahini, S.; Younesi, M.; Mortazavi, G.; Samare-Najaf, M.; Karim Azadbakht, M.; Jamali, N. Non-invasive diagnosis of endometriosis: Immunologic and genetic markers. Clin Chim Acta. 2023; 538,70-86.

5. Reviewer comment: The authors have not adequately highlighted the strengths and limitations of their study. I suggest better specifying these points.

Reply: According to the comment of the reviewer, we have extended the paragraph discussing the strengths of our study in the revised version of our manuscript. This paragraph now reads as follows:

‘Notably, these samples originated from transgenic FVB-Tg(CAG-luc-GFP)L2G85Chco/J mice and FVB/N wild-type mice, which enabled us to analyze the cellular exchange between individual lesions by means of bioluminescence imaging. Moreover, the samples were fixed by sutures in defined locations at the abdominal wall of the recipient animals, which allowed their easy retrieval for high-resolution ultrasound analyses [11]. Both imaging technologies could be performed repeatedly in a non-invasive manner, which markedly reduced the number of animals required for this study in line with the 3R principle. Moreover, high-resolution ultrasound imaging of endometriotic lesions allowed the separate quantification of their stromal and cyst volume, enabling a differentiated statement regarding their tissue proliferation and secretory activity.’

(See lines 271-280; marked in yellow)

In addition, we have included a novel paragraph in the revised version of our manuscript, which discusses the limitations of our study. This paragraph reads as follows:

‘Finally, it should be mentioned that the present study has also some limitations. In our model, we surgically induced endometriotic lesions by transplanting uterine tissue samples of healthy donor mice without the use of pathological endometriotic tissue of human origin. Therefore, the results obtained in this mouse model may not fully correlate to human patients with endometriosis. Moreover, due to ethical reasons the number of analyzed mice was reduced to a minimum, which may limit the statistical significance of our results. Furthermore, we analyzed the growth of newly developing endometriotic lesions only within 28 days. Hence, we cannot exclude that longer observation time periods would have shown different results.’

(See lines 331-339; marked in yellow)

6. Reviewer comment: The whole text should be corrected by a native English speaker in order to make the work clearer and more readable.

Reply: See also our reply to comment 1. According to the comment of the reviewer, we have again checked the manuscript for correct English language and typos.  

Round 2

Reviewer 1 Report

Thank you fotr authors for all the modifications

The paper can be accepted

Reviewer 3 Report

Thank you for going through the manuscript and the reviewers' points
In my honest opinion, the authors have responded satisfactorily to the reviewers’ criticisms.
The manuscript is well written and falls within the aim of this Journal.